# Self-Calibrating Magnetometer-Free Inertial Motion Tracking of 2-DoF Joints

**DOI:** 10.3390/s22249850

**Published:** 2022-12-15

**Authors:** Daniel Laidig, Ive Weygers, Thomas Seel

**Affiliations:** 1Control Systems Group, Technische Universität Berlin, 10623 Berlin, Germany; 2Department Artificial Intelligence in Biomedical Engineering, Friedrich-Alexander-Universität Erlangen-Nürnberg, 91052 Erlangen, Germany

**Keywords:** anatomical calibration, sensor-to-segment calibration, kinematic constraints, human motion analysis, elbow joint, inertial sensor, inertial measurement unit

## Abstract

Human motion analysis using inertial measurement units (IMUs) has recently been shown to provide accuracy similar to the gold standard, optical motion capture, but at lower costs and while being less restrictive and time-consuming. However, IMU-based motion analysis requires precise knowledge of the orientations in which the sensors are attached to the body segments. This knowledge is commonly obtained via time-consuming and error-prone anatomical calibration based on precisely defined poses or motions. In the present work, we propose a self-calibrating approach for magnetometer-free joint angle tracking that is suitable for joints with two degrees of freedom (DoF), such as the elbow, ankle, and metacarpophalangeal finger joints. The proposed methods exploit kinematic constraints in the angular rates and the relative orientations to simultaneously identify the joint axes and the heading offset. The experimental evaluation shows that the proposed methods are able to estimate plausible and consistent joint axes from just ten seconds of arbitrary elbow joint motion. Comparison with optical motion capture shows that the proposed methods yield joint angles with similar accuracy as a conventional IMU-based method while being much less restrictive. Therefore, the proposed methods improve the practical usability of IMU-based motion tracking in many clinical and biomedical applications.

## 1. Introduction

Marker-based optical motion capture (OMC) is considered the gold standard for human motion analysis. However, this method is time-consuming and confined to expensive laboratory environments. Ambulatory real-time motion analysis can be achieved at much lower costs with inertial measurement units (IMUs). Recent studies have shown that the accuracy of IMU-based motion analysis is comparable to marker-based OMC, see, e.g., [1,2].

However, in order to derive anatomically meaningful kinematic quantities, for example, joint angles, the orientation of each IMU with respect to its body segment must be known, as illustrated in Figure 1. Even small misalignments between the assumed and actual orientation of the IMUs on the body lead to errors in the obtained kinematic quantities. To ensure accurate motion tracking, it is therefore desirable to accurately determine this orientation. In practice, this is often achieved by manual placement of the IMUs on the respective body segments in a specified orientation [3], which is error-prone, especially when the attachment of sensors is to be performed by patients or by non-medical personnel.

An alternative is to include a procedure that determines the orientation of each IMU with respect to its body segment based on data measured by the sensors. This procedure is called *anatomical calibration* or *sensor-to-segment calibration*, which is not to be confused with sensor calibration. Sensor calibration determines parameters such as scaling and bias in order to increase the accuracy of the sensor orientation estimates. Anatomical calibration determines how the sensors are attached to the body segments to ensure that the rotation axes used for calculating joint angles match the anatomical axes of joint rotation.

As detailed in Section 2, anatomical calibration traditionally relies on precisely defined calibration poses or motions. Less restrictive approaches aim for anatomical calibration based on arbitrary joint motion. Such approaches have been proposed for (approximate) hinge joints [4,5]. In the following, we consider the more challenging case of joints with two degrees of freedom (DoF), such as the elbow joint (capable of flexion/extension and pronation/supination), the metacarpophalangeal joints (MCP) of the finger (capable of flexion/extension and adduction/abduction), or the ankle joint (capable of plantar-/dorsiflexion and inversion/eversion). The present contribution introduces methods for self-calibrating joint angle tracking that

Use two kinematic constraints for 2-DoF joints, one that must be fulfilled by the angular rates (as already introduced in [6,7]) and a novel constraint that must be fulfilled by the relative segment orientations at any time and for any motionDo not make use of magnetometer measurements and are therefore insensitive to magnetic disturbances (otherwise, temporary magnetic disturbances could permanently deteriorate accuracy until calibration is repeated)Instead simultaneously estimate the heading offset to facilitate magnetometer-free joint angle tracking.

The methods are evaluated based on two experiments. The first experiment, with a known sensor attachment as ground truth, compares a simple and a complex motion and is used to show that estimation over a short time window of just ten seconds of joint motion yields plausible and consistent joint axes. The second experiment, with OMC as ground truth, is used to validate that, while being much less restrictive, the proposed self-calibrating joint angle tracking provides the same accuracy as a conventional IMU-based approach.

## 2. Brief Review of the State of the Art in Anatomical Calibration

Anatomical calibration is the task of determining how the IMUs are attached to the body segments. In a broader sense, this also encompasses the pairing of IMUs to body segments [8,9] and the estimation of joint center positions [10,11,12,13]. The most relevant aspect, however, is to determine how the sensor coordinate system is rotated with respect to anatomical body segment axes (cf. Figure 1). In order to uniquely define this orientation, the coordinates of two anatomical axes need to be known in the sensor frame (or vice versa). Since errors in the sensor-to-segment orientations lead to kinematic cross-task and thus directly cause errors in the obtained joint angles [14,15,16], the reliability and accuracy of anatomical calibration methods are of fundamental interest in IMU-based motion analysis.

There are four main approaches for how to deal with the need for sensor-to-segment alignment in IMU-based human motion analysis [3]:Relying on a precisely defined sensor attachment (*assumed alignment*),Calibration via measurements from additional devices (*augmented data*),Calibration based on precisely defined poses or motions (*functional alignment*),Calibration from arbitrary motions (*model-based alignment*).

Using a precisely defined attachment of the sensors to the body is a common approach and, according to the survey by Vitali and Perkins [3], used by 42% of recent publications. The advantage of this approach is that it only requires minimum effort from the subject, i.e., no extra calibration movements are required, and that it is simple to implement. However, placing the sensors on the body so that predefined sensor axes correspond to functional joint axes is error-prone even for experienced medical personnel and even more so when patients themselves attach the sensors. In a study with three operators, Bouvier et al. [17] report reproducibility in the range of 4${}^{\circ }$to 12${}^{\circ }$ and agreement with OMC in the range of 8${}^{\circ }$to 23${}^{\circ }$.

An example of an augmented data method for anatomical calibration is the use of an additional custom device equipped with an IMU that is used to determine the sensor orientation with respect to anatomical landmarks [18,19].

The third approach is to ask the subject to assume precisely defined postures or perform a sequence of precisely defined motions. In the simplest form, this consists of a single pose calibration, often in the N-pose or T-pose [20,21,22,23], and requires magnetometers in order to be able to define two axes from one pose. A magnetometer-free alternative is to use two poses, e.g., one standing up and one lying down [24], or to derive the anatomical axes from angular rate measurements of precisely defined motions, typically around the functional axes of the joint [25,26,27]. Often, both approaches are combined, and one axis is derived from a static pose and one from a functional motion. Those hybrid approaches have been demonstrated for the upper body [28,29] and lower body [30,31,32]. For thorax and lumbar joint angles, however, a recent study by Cottam et al. [33] found that calibration via functional motions did not improve accuracy in comparison to relying on manual sensor placement. Bouvier et al. [17] observe similar accuracy for precise attachment and for various calibration approaches based on precise poses and motions and point out that accuracy depends more on the rigor of the experimental procedure and operator training than on the calibration method. Furthermore, performing those motions can be tedious for the subject, especially considering that a precise execution is required. For patients with motor disabilities, performing precise motions can be hard or impossible. Even after solving those obstacles, the main drawback of those methods is that the accuracy of the calibration depends on the accuracy of performing the motion. An elegant recent approach is to use the actual motions of interest for calibration, e.g., during cycling [34] or walking [35]. However, this is only feasible in a limited amount of applications and relies on strong assumptions on the analyzed motions.

In many cases, e.g., clinical applications, it would render the use of IMUs much more practical if both a precisely known attachment and precisely specified calibration poses and motions could be avoided by determining the sensor-to-segment orientations from arbitrary motions, usually by relying on kinematic constraints of biomechanical models. This was demonstrated for the knee joint by exploiting a kinematic constraint in the angular rates of (approximate) hinge joints [4,11]. Furthermore, it was shown that extending this constraint for a combined optimization of a three-segment chain improves robustness [36] and that other methods, such as principal component analysis [37] and factor graph optimization [38,39], can be used to exploit hinge joint constraints. In [5,40], the gyroscope-based hinge joint constraint introduced in [11] and an accelerometer-based constraint are combined with an elaborate sample selection strategy, and in [41], both constraints are analyzed for observability of the joint axis. Taetz et al. [42] introduce an approach based on sliding window weighted least squares optimization that uses hinge-joint and range-of-motion constraints and a body-shape prior to simultaneously estimate the sensor-to-segment orientation along with the body motion. Zimmermann et al. [9] demonstrate that deep learning can be used for lower body anatomical calibration with just two seconds of walking data.

For anatomical calibration based on arbitrary motions of 2-DoF joints, the existing work is limited. Müller et al. [6] introduce a gyroscope-based kinematic constraint for 2-DoF joints such as the elbow. Norden et al. [43] demonstrate that the same constraint can be employed for real-time estimation of hip and knee joint axes. However, the constraint used in both [6,43] assumes knowledge of the relative sensor orientation and therefore requires magnetometers. This poses a severe limitation for the applicability of those methods in indoor environments [44] and implies that temporary magnetic disturbances during calibration can lead to wrong axis estimates and thus permanently deteriorate the accuracy of the obtained joint angles. In [7], we presented first results of a magnetometer-free method that overcomes those restrictions by simultaneously estimating the heading offset.

## 3. Kinematic Model of 2-DoF Joints

The methods proposed in the present contribution perform automatic anatomical calibration for joints with two degrees of freedom (DoF). Those methods are suitable for any 2-DoF joint and can be applied to a range of biomechanical or robotic 2-DoF joints. To improve comprehensibility, the following description of the kinematic model and the calibration method focuses on the human elbow joint as an exemplary joint, which is later also used in the experimental evaluation.

Furthermore, even though in the following we always only consider two body segments connected by a single joint, the proposed methods can be used to analzye longer kinematic chains consisting of multiple segments. In this case, the calibration methods can be applied to each pair of segments that are connected by a 2-DoF joint.

Figure 2 shows an anatomical model of the elbow joint as an exemplary biological 2-DoF joint. This joint can perform two functional motions. Flexion and extension (FE) are performed by the humeroulnar joint, while pronation and supination (PS) are the result of the radius pivoting around the ulna.

As an approximation, we can model this joint—as well as any other 2-DoF joint—as a kinematic chain consisting of two hinge joints and one fixed rotation in between, as depicted in Figure 3. Including the fixed rotation, the sequence of rotations consists of flexion and extension (FE), a fixed carrying angle [45], and pronation and supination (PS).

We use unit quaternions to denote rotations and orientations [46]. In the context of quaternion multiplication, which we denote by ⊗, we implicitly regard 3D vectors as pure quaternions. Square brackets specify the coordinate system in which a vector is expressed, for example, ${\left[\omega_{1}\right]}_{E}$ is the gyroscope measurement of IMU $S{}_{1}$ transformed into frame E, i.e., [ω1]E=qES⊗ω1⊗qES−1. Here, the left upper and lower indices denote the frames between which the quaternion rotates. Quaternions that represent the rotation of an angle α∈R around the axis $v_{}\in R^{3}$ are written as $\left(\alpha \;@\;v_{}\right):={\left[\begin{array}{cc} \cos\frac{\alpha }{2} & \frac{v_{}^{⊺}}{∥v_{}∥}\sin\frac{\alpha }{2} \end{array}\right]}^{⊺}$.

We can use this notation to mathematically express the orientation of the forearm $B{}_{2}$ relative to the upper arm $B{}_{1}$ using the FE joint angle α(t), the carrying angle $\beta_{0}$, and the PS angle γ(t) as
(1)qB1B2=α(t)@j1⊗β0@j1×j2⊗γ(t)@j2.

The International Society of Biomechanics (ISB) [45] also recommends this joint model for the elbow and precisely defines coordinate systems $B{}_{1}$ and $B{}_{2}$ so that ${\left[j_{1}\right]}_{B1}={\left[0\;0\;1\right]}^{⊺}$ and ${\left[j_{2}\right]}_{B2}={\left[0\;1\;0\right]}^{⊺}$. When using this definition, the joint angles are intrinsic $z-x^{\prime }-y^{\prime \prime }$ Euler angles of qB1B2. Please note that this also means that the axis $j_{1}$ (FE) is fixed in the coordinate system of a sensor attached to the upper arm, while the axis $j_{2}$ (PS) is fixed in the coordinate system of a sensor attached to the forearm.

Instead of using regular Euler angles, we could consider modeling a 2-DoF joint with axes that are all potentially non-orthogonal (including the carrying angle axis). However, as Appendix A shows, any generic model with non-orthogonal axes can also be expressed using standard $z-x^{\prime }-y^{\prime \prime }$ Euler angles by redefining the segment coordinate systems accordingly. This means that the choice of $z-x^{\prime }-y^{\prime \prime }$ Euler angles according to the ISB recommendations [45] does not restrict the generality of the proposed methods. Moreover, note that the orientation of the IMUs on the body segments is independent of this definition. The goal of anatomical calibration is to determine the fixed coordinates $j_{1}$ and $j_{2}$ of the functional joint axes in the local coordinate systems of the respective IMUs.

## 4. Proposed Methods

Two IMUs $S{}_{1}$ and $S{}_{2}$ are placed on the subject in unknown orientations, one on each body segment connected by the 2-DoF joint (i.e., in case of the elbow, one on the upper arm and one on the forearm). Assume that we can estimate the sensor orientation quaternions qES1(tk), qES2(tk) relative to a common inertial frame E. We also measure the angular rates $\omega_{1}\left(t_{k}\right)\in R^{3}$, $\omega_{2}\left(t_{k}\right)\in R^{3}$ of the IMUs, in their respective local coordinate systems. All measurements are sampled at times $t_{k}=kT_{s},k\in \left\{1,2,\ldots ,N\right\},T_{s}\in R_{>0}$. Note that the assumption of a common inertial frame E is restrictive in practice as it assumes 9D sensor fusion in a perfectly homogeneous magnetic field and will later be dropped.

In the following, we derive two different kinematic constraints for 2-DoF joints, one based on the joint rotation and one based on the relative segment orientations. Both constraints are suitable for 6D sensor fusion with unknown heading offset. Given a short sequence of recorded IMU data, we use the Gauss–Newton algorithm to determine the joint axes coordinates in the sensor frame and the heading offset that best fit either the rotation-based or the orientation-based constraint in a least-squares sense. We use these joint axes coordinates to determine segment orientations from the sensor orientations and use the heading offset to align the reference frames of the orientations. From this result, we calculate the relative segment orientation, which we then decompose via Euler angles to obtain magnetometer-free estimates of the joint angles.

### 4.1. Rotation-Based Kinematic Joint Constraint

As shown in Section 3, a 2-DoF joint cannot perform arbitrary joint rotation in all directions. Instead, rotation is only possible around the two joint axes. In the following, we will investigate how this translates to a kinematic constraint in the angular rates measured by the two IMUs. We will later exploit this constraint to estimate joint axes from arbitrary joint motion.

Using the addition theorem for angular velocities, we express the relationship between the gyroscope measurements $\omega_{1}\left(t_{k}\right)$ and $\omega_{2}\left(t_{k}\right)$ as
(2)$${\left[\omega_{2}\right]}_{E}={\left[\omega_{1}\right]}_{E}+\omega_{j_{1}}{\left[j_{1}\right]}_{E}+\omega_{j_{2}}{\left[j_{2}\right]}_{E}.$$

The scalars $\omega_{j_{1}}$ and $\omega_{j_{2}}$ are the rotation rates of the joint around the respective joint axes. In case of joints with two degrees of freedom according to the model in Figure 3, this corresponds to the anatomical joint motions, i.e., in case of the elbow, $\omega_{j_{1}}$ is the FE angular rate and $\omega_{j_{2}}$ the PS angular rate. This means that the angular rate $\omega_{2}$ measured by the forearm IMU $S{}_{2}$ is composed of three components:The common rotation of the whole arm, also observed by IMU $S{}_{1}$ as $\omega_{1}$The FE rotation around $j_{1}$The PS rotation around $j_{2}$.

Note that the carrying angle does not appear, since it is time-invariant. Furthermore, note that in (Equation 2), the angular rates and joint axes are transformed into a common coordinate system, here E.

For hinge joints, in [4], the following constraint has been derived from (Equation 2):(3)$$∥\omega_{1}\times j_{1}∥-∥\omega_{2}\times j_{2}∥=0.$$

Since this version of the constraint only uses quantities given in local sensor coordinates, it is independent of sensor orientations with respect to a fixed frame and thus not affected by magnetic disturbances.

For joints with two degrees of freedom, we need to know the relative sensor orientation or sensor orientations with respect to a common fixed frame. In order to derive a similar constraint from (Equation 2) for 2-DoF joints, we calculate the scalar product with the normalized axis ${\left[j_{1}\right]}_{E}\times {\left[j_{2}\right]}_{E}$ on both sides, i.e.,
(4)$$\left({\left[\omega_{2}\right]}_{E}-\omega_{j_{2}}{\left[j_{2}\right]}_{E}\right)\;\cdot \;\frac{{\left[j_{1}\right]}_{E}\times {\left[j_{2}\right]}_{E}}{∥{\left[j_{1}\right]}_{E}\times {\left[j_{2}\right]}_{E}∥}=\left({\left[\omega_{1}\right]}_{E}+\omega_{j_{1}}{\left[j_{1}\right]}_{E}\right)\;\cdot \;\frac{{\left[j_{1}\right]}_{E}\times {\left[j_{2}\right]}_{E}}{∥{\left[j_{1}\right]}_{E}\times {\left[j_{2}\right]}_{E}∥},$$
and employ the fact that $a_{}\;\cdot \;\left(a_{}\times b_{}\right)=a_{}\;\cdot \;\left(b_{}\times a_{}\right)=0$. This yields
(5)$$\left({\left[\omega_{1}\right]}_{E}-{\left[\omega_{2}\right]}_{E}\right)\;\cdot \;\frac{{\left[j_{1}\right]}_{E}\times {\left[j_{2}\right]}_{E}}{∥{\left[j_{1}\right]}_{E}\times {\left[j_{2}\right]}_{E}∥}=0.$$

Note that normalizing the axis was found to improve robustness compared to the constraint presented in [47].

For perfect 2-DoF joints and ideal IMU measurements, this constraint must be fulfilled for each sampling instant. For biological joints, and when taking soft tissue motion and measurement errors into account, the constraint is still valid in a least-squares sense when considering a short motion sequence consisting of multiple samples.

However, the constraint as formulated in (Equation 5) uses the reference frame E and is only suitable for use in combination with 9D inertial orientation estimation, i.e., with the use of magnetometers. Since magnetic fields are often severely disturbed [44], we want to avoid using magnetometer measurements and therefore only employ 6D sensor fusion to estimate the sensor orientations, e.g., using the VQF algorithm [48]. This implies that the heading of the estimated orientations is not well-defined. More precisely, this can be described by the estimated orientations qE1S1 and qE2S2 being given in different global reference frames $E{}_{1}$ and $E{}_{2}$, which are rotated around the vertical global *z*-axis, i.e.,
(6)qE1E2=δ(t)@0 0 1⊺=cosδ(t)200sinδ(t)2⊺.

The heading offset δ(t) has an unknown initial value and then slowly drifts due to gyroscope bias [49]. Please note that both $E_{1}$ and $E_{2}$ have some unknown heading offset with respect to a fixed frame E used in 9D sensor fusion and defined by gravity and the Earth’s magnetic field. However, knowing those individual offsets is not necessary for calculating relative orientations and joint angles.

We take the heading offset into account by evaluating the constraint (Equation 5) in one of the slowly-drifting global frames (here $E{}_{1}$), i.e.,
(7)$$\underset{=:\;\omega_{\mathrm{rel}}}{\underset{︸}{\left({\left[\omega_{1}\right]}_{E{}_{1}}-{\left[\omega_{2}\right]}_{E{}_{1}}\right)}}\;\cdot \;\underset{=:\;j_{n}/∥j_{n}∥}{\underset{︸}{\frac{{\left[j_{1}\right]}_{E{}_{1}}\times {\left[j_{2}\right]}_{E{}_{1}}}{∥{\left[j_{1}\right]}_{E{}_{1}}\times {\left[j_{2}\right]}_{E{}_{1}}∥}}}=0.$$

This version of the constraint implicitly depends on δ, as we need the quaternion qE1S2=qE1E2(δ)⊗qE2S2 to transform $\omega_{2}$ and $j_{2}$ to $E_{1}$ coordinates. This means that instead of (Equation 5) we can use (Equation 7) with magnetometer-free 6D orientations and that, in addition to the joint axes coordinates, we also identify the current heading offset δ(t) as an additional parameter.

### 4.2. Orientation-Based Kinematic Joint Constraint

As an alternative, we derive a second kinematic joint constraint. In contrast to the constraint introduced in the previous section, this constraint is not based on the joint rotation but on the joint orientation, i.e., the relative orientation between the two body segments connected by the joint.

As in Section 4.1, assume that we have 6D sensor orientation estimates qE1S1(tk), qE1S2(tk), e.g., estimated with the VQF algorithm [48]. As before, our aim is to identify ${\left[j_{1}\right]}_{S{}_{1}}$, ${\left[j_{2}\right]}_{S{}_{2}}$, and the heading offset δ(t). For any given estimate of those values, we are able to calculate joint angles. If the joint follows the 2-DoF joint model introduced in Section 3, the following statement holds true: With the correct sensor-to-segment orientation and the correct heading offset, the second joint angle (for the elbow joint: the carrying angle) is constant.

Mathematically, we can formulate this by calculating the joint orientation and then decomposing this orientation into Euler angles. First, we determine the shortest-possible rotations that align the estimated sensor axes with the joint axes: (8)qS1B1=arccos0 0 1⊺·j1S1@0 0 1⊺×j1S1(9)qS2B2=arccos0 1 0⊺·j2S2@0 1 0⊺×j2S2
and calculate the rotation quaternion between the reference frames
(10)qE1E2=δ@0 0 1⊺.

Using those quaternions we calculate the joint orientation
(11)qB1B2=qB1S1⊗qS1E1⊗qE1E2⊗qE2S2︸=qS1S2⊗qS2B2,
which depends on the sensor orientations, the estimated joint axes $j_{1}$ and $j_{2}$, and the heading offset δ.

Therefore, qB1B2=:qw qx qy qz⊺ can be calculated from the measured data and the estimated parameters. The second intrinsic $z-x^{\prime }-y^{\prime \prime }$ Euler angle of this quaternion, i.e., the estimated carrying angle, is
(12)$$\begin{array}{c} {\hat{\beta }}_{0}=\arcsin\left(2q_{w}q_{x}+2q_{y}q_{z}\right). \end{array}$$

Due to the joint constraint, this angle has to be constant over the whole measurement window, i.e., with the fixed constant carrying angle $\beta_{0}$,
(13)$$\begin{array}{c} \arcsin\left(2q_{w}q_{x}+2q_{y}q_{z}\right)=\beta_{0}. \end{array}$$

Similar to (Equation 7), the constraint (Equation 13) can be used to identify the joint axes coordinates and the heading offset δ. Additionally, unless the actual value of the carrying angle $\beta_{0}$ is known, $\beta_{0}$ has to be identified as an additional parameter.

### 4.3. Parametrization of Joint Axes

The aim of the anatomical calibration is to identify the joint axes $j_{1}\in R^{3}$ and $j_{2}\in R^{3}$ with $∥j_{i}∥=1$, i=1,2. Parametrizing the axes as Cartesian vectors in an optimization problem is inconvenient as we would need an additional constraint to ensure unit length. Therefore, we employ spherical coordinates and represent each axis by two parameters $\varphi_{i}$ and $\theta_{i}$, e.g.,
(14)$$j_{i}={\left[\sin\theta_{i}\cos\varphi_{i}\;\sin\theta_{i}\sin\varphi_{i}\;\cos\theta_{i}\right]}^{⊺},\;i=1,2.$$

With the parametrization given in (Equation 14), $\frac{\partial j_{i}}{\partial \varphi_{i}}=0$ if $\sin\theta_{i}=0$. To avoid this singularity, we introduce an alternative spherical representation of the same joint axis direction, as shown in Figure 4. During optimization, we always use a parametrization with $|\sin\theta_{i}|\gg 0$ by converting the axis to Cartesian coordinates and then to the other representation whenever the current representation comes close (<30°) to that singularity.

Note that both spherical parametrizations represent exactly the same 3D vector. Therefore, changing the parametrization in between optimization iterations does not influence the joint axis vectors or the value of the cost function but ensures that the derivatives with respect to the joint axes are always sufficiently sensitive.

### 4.4. Cost Function and Optimization

Sample selection is performed to fill a sample buffer of *M* data sets
(15)qE1S1(tk),qE2S2(tk),ω1E1(tk),ω2E2(tk)
for the rotation-based constraint and
(16)qE1S1(tk),qE2S2(tk)
for the orientation-based constraint from the 6D orientation quaternions and angular rates measured at a (potentially very high) sampling frequency of $f_{s}$. The proposed method employs a regular (equidistant) sample selection strategy that stores one sample every 0.05 s. Note that this method can easily be extended by more sophisticated sample selection strategies since the optimization procedure does not require equidistant sampling.

In order to determine the joint axes and heading offset that best satisfy the rotation-based constraint (Equation 7) in a least-squares sense, we define the error for each sampling instant $t_{k}$ as
(17)$$e\left(t_{k}\right):=\omega_{\mathrm{rel}}\left(\delta \right)\;\cdot \;\frac{j_{n}\left(\Phi_{}\right)}{∥j_{n}\left(\Phi_{}\right)∥},$$
with the parameter vector $\Phi_{}:={\left[\theta_{1}\;\varphi_{1}\;\theta_{2}\;\varphi_{2}\;\delta \right]}^{⊺}$. Note that we assume the heading offset δ(t) to be constant for all samples in the current buffer, which is valid for short window lengths.

Similarly, for the orientation-based constraint (Equation 13), we define the error as
(18)$$e\left(\Phi_{}\right):=\arcsin\left(2q_{w}q_{x}+2q_{y}q_{z}\right)-\beta_{0},$$
with a parameter vector $\Phi_{}:={\left[\theta_{1}\;\varphi_{1}\;\theta_{2}\;\varphi_{2}\;\delta \;\beta_{0}\right]}^{⊺}$ that additionally includes the carrying angle.

To estimate the joint axes $j_{1}$ and $j_{2}$ and the heading offset δ given a set of *M* samples, we find the parameter vector $\hat{\Phi_{}}$ that minimizes the error of either the rotation-based constraint or the orientation-based constraint using the Gauss–Newton algorithm [50]. Appendix B gives details on the optimization algorithm, provides analytical expressions for the gradients of the cost functions, and introduces a moving window approach for employing the proposed method in real-time applications. As a result of the optimization step, we obtain the joint axes $j_{1}$ and $j_{2}$ in the coordinates systems of sensors $S{}_{1}$ and $S{}_{2}$, respectively, and the heading offset δ between the reference frames $E{}_{1}$ and $E{}_{2}$.

### 4.5. Joint Angle Calculation

Using the optimization results, we calculate FE and PS joint angles based on the ISB recommendations [45]. Those joint angles are defined as intrinsic $z-x^{\prime }-y^{\prime \prime }$ Euler angles of the forearm $B{}_{2}$ relative to the upper arm $B{}_{1}$, i.e., qB1B2, with $B{}_{1}$ and $B{}_{2}$ being the segment coordinate systems as defined in [45].

From 6D inertial orientation estimation, we obtain the sensor orientation quaternions qE1S1 and qE2S2. After performing the optimization, we know the coordinates of both joint axes $j_{1}$ and $j_{2}$ in local sensor coordinates and the heading offset δ. Note that additional knowledge is needed to determine the absolute value of the joint angles without any offset—for example, for the elbow joint, which joint orientation corresponds to zero flexion and zero pronation is only a matter of convention and not an inherent property of the 2-DoF joint. To obtain offset-free angles, we employ reference values of the FE and PS joint angles at one arbitrary time instant $t_{\mathrm{ref}}$, e.g., obtained from a known pose or by exploiting the maximum range of motion of the joint. With those values, the joint angles can be calculated by the algorithm described below:

First, we calculate qE1E2 via (Equation 6) and use this to obtain qE1S2=qE1E2⊗qE2S2. Then we determine rotations that ensure that the identified joint axes match the joint axes defined in [45]:(19)qS1B1′=arccos([001]⊺·j1)@[001]⊺×j1(20)qS2B2′=arccos([010]⊺·j2)@[010]⊺×j2.

Using those, we calculate the relative segment orientation
(21)qB1′B2′=qE1S1⊗qS1B1′−1⊗qE1S2⊗qS2B2′.

For any quaternion $q_{}=:{\left[q_{w}\;q_{x}\;q_{y}\;q_{z}\right]}^{⊺}$, the $z-x^{\prime }-y^{\prime \prime }$ Euler angles (α,β,γ) can be calculated as
(22)$$\begin{array}{cc} \alpha & =atan2(2q_{w}q_{z}-2q_{x}q_{y},q_{w}^{2}-q_{x}^{2}+q_{y}^{2}-q_{z}^{2}), \end{array}$$
(23)$$\begin{array}{cc} \beta & =\arcsin(2q_{w}q_{x}+2q_{y}q_{3}), \end{array}$$
(24)$$\begin{array}{cc} \gamma & =atan2(2q_{w}q_{y}-2q_{x}q_{z},q_{w}^{2}-q_{x}^{2}-q_{y}^{2}+q_{z}^{2}). \end{array}$$

By calculating $z-x^{\prime }-y^{\prime \prime }$ Euler angles $\left(\alpha^{\prime },\beta^{\prime },\gamma^{\prime }\right)$ of qB1′B2′, we obtain the FE angle $\alpha^{\prime }$ and the PS angle $\gamma^{\prime }$ that only differ from the well-defined joint angles according to [45] by a constant offset that depends on the actual placement of the IMUs.

We can eliminate this offset by exploiting knowledge of the actual joint angles $\alpha_{\mathrm{ref}}$ and $\gamma_{\mathrm{ref}}$ at $t=t_{\mathrm{ref}}$. The segment-to-sensor orientations
(25)qS1B1=qS1B1′⊗α′(tref)−αref@[001]⊺,
(26)qS2B2=qS2B2′⊗γref−γ′(tref)@[010]⊺
allow us to calculate qB1B2=(qE1S1⊗qS1B1)−1⊗qE1S2⊗qS2B2. The Euler angles $\left(\alpha ,\beta_{0},\gamma \right)$ of qB1B2 are the offset-free FE and PS joint angles α and γ, respectively, and the carrying angle $\beta_{0}$ (cf. Figure 3), which is almost constant and rarely reported [45].

To further improve the proposed method, in Appendix C, we introduce an optional extension that allows for the rotation-based constraint to be used when only orientation data are available (e.g., if on-chip sensor fusion is used), add a low-pass filter to reduce the influence of soft tissue motion artifacts, and discuss options for how to resolve the ambiguity in the signs of the joint axes.

## 5. Experimental Evaluation

We evaluate the proposed magnetometer-free anatomical calibration and joint angle calculation methods based on two experiments.

The first experiment is designed to evaluate if the obtained joint axis estimates are plausible and consistent. To this end, IMU data from two different motions are recorded from five subjects and a mechanical joint, while carefully attaching the sensors in a known orientation. Each trial is split into overlapping time windows to which the anatomical calibration methods are applied. The obtained joint axis estimates are compared to the axes obtained by the more restrictive method of careful manual sensor placement.

The second experiment is designed for the evaluation of the accuracy of the obtained joint angles with the proposed self-calibrating magnetometer-free joint angle calculation method. This experiment consists of recordings of natural everyday life motions of two subjects. It uses marker-based OMC as a reference, which allows for the comparison of the obtained joint angles to joint angles obtained from optical markers and from a conventional 9D IMU-based approach. As a further validation step, we consider the variability of the expected-to-be-constant carrying angle as a metric for how well the estimated joint axes describe the functional joint motion.

Note that in all experiments, the sensors are carefully attached in a known orientation to facilitate a plausibility check of the obtained results. To still verify that the proposed methods do not make assumptions regarding the sensor orientation, we simulate a random sensor attachment by multiplying all gyroscope and accelerometer measurements with a random rotation matrix that is different for each time window.

The extension for on-chip sensor fusion introduced in Appendix C. is always used, i.e., the angular rates used for evaluating the rotation-based kinematic constraint are derived from the orientation estimates. Since the impact on the results is negligible, the results obtained using the actual gyroscope measurements are not shown separately.

### 5.1. Robustness of Joint Axis Estimation

The first experiment is performed to answer the following two research questions:Are the estimated joint axes plausible, i.e., do they agree with the values expected based on careful manual placement?Are the estimated joint axes consistent, i.e., do we always obtain the same result when using different parts of the trial?

Data from five healthy subjects is recorded. The subjects are adult volunteers with no history of upper-limb injury that might affect upper-limb movement. Inertial sensors (Xsens MTw, Xsens Technologies B.V., Netherlands) are placed on the upper arm close to the elbow and on the forearm close to the wrist. The sensors are placed in a defined orientation on the skin so that one local sensor axis coincides roughly with the functional joint axis.

We define two different motions:The *simple motion* consists of FE of the elbow and PS of the forearm, performed alternatingly while keeping the longitudinal axes of upper arm and forearm parallel to the sagittal plane.For the *complex motion*, we ask the subject to perform random combinations of FE and PS, allowing for 3D rotation of the shoulder including humeral rotation.

Each subject performs both motions for approximately one minute.

In addition to the five human subjects, an additional data set is recorded using a mechanical joint. This joint has dimensions similar to the human arm and consists of two hinge joints as shown in Figure 3. During the recordings, the joint was held in hand and moved in a way that mimics the motions performed by the five subjects.

For each recording, the proposed methods are used on 21 partially overlapping moving windows *w*, w=1,2,…,21, of length 10s with data sets recorded every 0.05 s. Note that we will later investigate the effect of window length and sampling time and show that this window length is usually sufficient to identify the joint axes and that collecting data sets more frequently does not significantly improve the robustness.

Since the only available ground truth are approximate axis coordinates that we know due to the orientation in which the sensor was attached, we define suitable evaluation metrics that allow us to quantify the consistency and plausibility of the estimates. See Figure 5 for an illustration of the definitions. First, denote the estimated joint axes $j_{w}$, with *w* being the index for the estimation window. For a compact notation, we omit the segment index, denoting whether the axis is a flexion and extension (FE) axis or a pronation and supination (PS) axis. To assess if the estimates are consistent, we define the *variability angle*
(27)$$\varepsilon_{w}=∢\left(j_{w},\;j_{\mathrm{mean}}\right),$$
where *∢* denotes the positive angle between two 3D vectors and
(28)$$j_{\mathrm{mean}}=\frac{1}{21}\sum_{w=1}^{21}j_{w}$$
is the mean of all estimates. In other words, $\varepsilon_{w}$ is the angular deviation between the estimate for window *w* and the mean of all estimates. If this angle is always small, the estimation results agree well for all time windows.

To also check if this result is plausible, we introduce the *misalignment angle*
(29)$$\alpha =∢(j_{\mathrm{mean}},\;j_{\mathrm{att}}),$$
with $j_{\mathrm{att}}$ being the joint axis obtained via careful manual sensor attachment. Therefore, α is the angle between the mean estimation result and the axis obtained via manual sensor attachment. While precise manual sensor attachment is hard and error-prone, we can at least expect both axes to coincide roughly and therefore consider the result plausible if α≤30°.

Figure 6 shows the results obtained in the first experiment with the rotation-based and orientation-based constraints. In general, we see that the proposed methods for anatomical calibration produce good results: with both constraints, the methods are able to determine plausible FE and PS joint axes from 10-second recordings, and in all cases except for Subject 2 with the orientation-based constraint and the complex motion, the median of the variability angle $\varepsilon_{w}$ is below 10°. In other words, almost all time windows lead to axis estimates within the expected range. As a main result, it is noticeable that the rotation-based constraint performs better than the orientation-based constraint and that a slight increase in the variability angles $\varepsilon_{w}$ can be observed in the complex motion. This is likely due to soft tissue motion caused by humeral rotation. Furthermore, the randomness of the complex motion can lead to longer periods of motion that do not excite both degrees of freedom of the joint.

The results obtained with the mechanical joint agree very well with the expected axes (α≤2°), and the joint axis estimates are more consistent than for the biological elbow joints. This is to be expected since precisely attaching the sensors is easier with the mechanical joints, there are no soft tissue motion artifacts, and the mechanical joint constructed with two hinge joints follows the kinematic model (Figure 3) more precisely than the human elbow.

To facilitate an intuitive understanding of the results, Figure 7 shows the estimated and expected joint axes in a 3D visualization of the respective IMU coordinate systems. We can see that, for both FE and PS, the joint axis estimates of all overlapping time windows agree well. While the PS axis agrees very well with the axis expected due to sensor alignment, a systematic disagreement of ∼17° between the estimated and expected axes is noticeable. Since all estimates are very consistent, this is most likely due to an imprecise manual attachment of the sensor, causing the *y*-axis to disagree with the functional PS axis of the joint. In general, we see in Figure 6 that the misalignment angle α is larger for the FE axis $j_{1}$ than for the PS axis $j_{2}$. This is plausible, given the fact that the longitudinal *x*-axis of the IMU is much easier to precisely align with the longitudinal axis of the forearm, whereas aligning the *y*-axis of the upper arm IMU, corresponding to a much shorter dimension of the sensor case, with the functional FE axis was found to be much harder while conducting the experiments.

However, it is noticeable that also for the variability angle $\varepsilon_{w}$, the values are typically much larger for the FE axis than for the PS axis, indicating that it is not only harder to perform a precise manual alignment of this axis but it is also harder for the proposed methods to accurately and consistently estimate this axis. This effect is especially pronounced for the complex motion.

To investigate one potential effect, we take a closer look at Subject 2 and the rotation-based constraint. In the complex motion trials, Subject 2 stands out as the range of motion of the upper arm IMU is significantly lower than for the other subjects (more specifically, the mean pairwise orientation difference within a window is 16° for Subject 2 and between 46° and 56° for the other four subjects) while the FE axis deviations are larger than for all other subjects. In Figure 8, we visualize the estimated FE joint axes (Figure 8a) and notice that all estimates lie approximately within the *y*-*z*-plane of the sensor. During the trial, the *x*-axis of the upper arm IMU was approximately vertical, i.e., the *y*-*z*-plane is approximately horizontal. When calculating the angle of the joint axis in this *y*-*z*-plane and plotting this angle together with the estimated heading offset δ in Figure 8b, we notice that there is an obvious correlation.

This correlation can be explained when considering the kinematic constraint in (Equation 7) for the special case in which the upper arm does not move, i.e., the orientation qE1B1 is constant, $\omega_{{}_{1}}=0$, and the coordinates of ${\left[j_{1}\right]}_{E{}_{1}}$ are constant. In this case, there is no difference between a change in δ, i.e., the heading offset between $E{}_{1}$ and $E{}_{2}$, and a rotation of the joint axis estimate $j_{1}$ around the vertical axis. The observation in Figure 8 is likely caused by the real situation being too close to this singular case. To mitigate this, care should be taken to avoid calibration motions during which one of the body segments is always stationary.

In summary, the evaluation of the first experiment has shown that the proposed methods yield consistent and plausible joint axis estimates. The rotation-based constraint performs better than the orientation-based constraint. To ensure that the axes converge, the subject’s motion should include sufficient motion from both the upper arm and the forearm.

To further enrich the evaluation, we use the data from the first experiment to investigate the influence of the the cutoff frequency for the low-pass filter, the sample selection frequency, and the window duration. The results are presented in Appendix D.

### 5.2. Accuracy of Magnetometer-Free Joint Angle Tracking

The second experiment is performed to validate that the proposed methods can be used to obtain accurate elbow joint angles for functional motions without relying on a precisely known sensor attachment and without relying on the magnetic field. An optical motion capture system (Vicon Motion Systems Ltd. UK) is used as reference. In addition to the two inertial sensors positioned as in the previous experiment, optical markers are placed on bony landmarks in a way that facilitates joint angle measurement as recommended by the ISB [45]. Note that by placing reflective markers on anatomical landmarks and not, like many previous works, on the IMUs, we ensure that we compare against the gold standard for measuring joint angles, taking soft tissue motion into account.

Two healthy adult volunteers, with no history of upper-limb injury that might affect upper-limb movement, performed two motions:During the *pick-and-place* motion, the subject placed a small box in a sequence of predefined orientations and locations on a table.The *drinking* motion consists of the subject repeatedly placing the hand on a table, grabbing a cup, simulating a drinking motion, and then placing the cup back on the table.

Each of the two subjects repeats the two motions four times (twice slow and twice fast), resulting in a total of 16 trials, with durations between 14 and 44 s.

For each trial, calculate four different joint angles.

The OMC-based *ground truth* angles are derived from the optical markers placed on anatomical landmarks and calculated as described in [45].*Conventional* IMU-based joint angles are calculated using 9D sensor fusion (with the VQF algorithm [48]), i.e., using the magnetic field to determine the heading, and relying on the careful placement of the sensors on the body.In contrast, the *proposed* IMU-based joint angles use 6D sensor fusion (with the VQF algorithm [48]), and the joint axes and heading offset are identified from the trial motion using the*rotation-based* joint constraint and the*orientation-based* joint constraint.

Note that the application of the proposed methods tests the most challenging case, i.e., we use a standard everyday motion to identify both the joint axes and the heading offset without requiring a separate calibration phase.

To determine the sign and the required offset for the joint angles, we use the OMC-based angles. The IMU-based joint angles obtained by the different methods are compared to the OMC-based ground truth, and the RMSE is calculated. Results from all trials are shown in Figure 9.

When comparing the two variants of the proposed method, we see that the rotation-based constraint outperforms the orientation-based constraint. This coincides with the results of the first experiment presented in Section 5.2. It is noteworthy that for many trials the accuracy achieved with both constraints is comparable and the difference in the mean accuracy is caused by several outliers obtained with the orientation-based constraint, which is consistent with the lower robustness observed for this constraint in Figure 6.

However, when considering the results obtained with the proposed method and the rotation-based constraint, the accuracy is similar to the conventional 9D IMU-based method. For the FE angles, the mean RMSE of 2.1° is 0.2° lower than for the conventional method, while for the PS angles, the mean RMSE of 3.7° is 0.1° larger. In contrast to the results with the orientation-based constraint, there are no outliers, and the maximum RMSE of the proposed method and the conventional method is comparable. Note that the conventional method relies on properly calibrated magnetometer measurements, a controlled environment without ferromagnetic material or electric devices, and a precise and known sensor attachment and is therefore much more restrictive than the proposed magnetometer-free plug-and-play method.

To illustrate the performed motions and the obtained results, Figure 10 shows the OMC ground truth joint angles, the conventional IMU-based joint angles, and the proposed joint angles with the rotation-based constraint for two exemplary trials. In several short time periods, ground truth data are not available due to occlusion, i.e., at least one of the required markers could not be tracked by the OMC system. Those phases were excluded from the RMSE calculation. As can be seen, the joint angles obtained with the proposed plug-and-play method agree well with both the conventional IMU-based joint angles and the OMC-based ground truth angles.

Note that the joint constraint is only used for identifying the joint axes and that the joint angle calculation uses standard Euler angles and therefore not directly restricted by this constraint. As a result, the obtained carrying angles, which are also shown in Figure 10 but rarely reported in practice, are not perfectly constant.

We can use the carrying angle as an indicator of how well the measured joint motion adheres to the 2-DoF joint model (Figure 3). For a perfect 2-DoF joint, we would expect a perfectly constant carrying angle, while a 3-DoF joint will show significant movement in all three joint angles. Furthermore, if the joint is in fact a 2-DoF joint but the joint axis estimates are wrong, the Euler decomposition will cause variability in all three joint angles.

Therefore, we calculate the standard deviation of the carrying angle as a measure of variability, which is shown in Figure 11 for all 16 trials and all four angle calculation methods. With both constraints, the median of the standard deviations is slightly lower than for the conventional IMU-based joint angles and the OMC-based ground truth. This indicates that the joint axis estimates automatically obtained with the proposed method are better suited to describe the functional motion of the joint than the axes obtained via careful IMU placement and the axes obtained via the placement of optical markers on anatomical landmarks. This agrees with previous research showing that anatomical joint axes defined based on anatomical landmarks do not coincide with the rotation axes of functional joint motion [51]. For joint angle calculation, the use of functional rotation axes seems preferable in order to minimize kinematic cross-talk.

In summary, the evaluation of the second experiment has shown that for the challenging case of using recordings of everyday motions for calibration, the proposed methods are able to obtain joint angles with the same accuracy as a conventional IMU-based approach, while not relying on precise sensor placement or magnetometer measurements. As also shown via the first experiment, the rotation-based constraint performs better than the orientation-based constraint and should therefore be used for anatomical calibration.

## 6. Conclusions

The present contribution introduced methods for automatic anatomical calibration for 2-DoF joints, such as the elbow, that do not require the subject to perform precise calibration movements but instead work on arbitrary motions by exploiting one of two kinematic constraints: a rotation-based constraint for the angular rates and an orientation-based constraint. The methods do not make use of magnetometer measurements. Instead, the heading offset is simultaneously estimated via the kinematic constraint, which facilitates plug-and-play magnetometer-free joint angle estimation.

The proposed methods were evaluated using two experiments. The first experiment, without OMC ground truth, showed that the proposed methods yield consistent and plausible joint axis estimates from only ten seconds of motion data. The second experiment, performed with OMC as ground truth, showed that the proposed plug-and-play method can estimate accurate joint angles while being much less restrictive than a conventional IMU-based approach. In both experiments, the rotation-based joint constraint performed better than the orientation-based joint constraint.

The proposed methods overcome mounting and calibration restrictions and facilitate magnetometer-free motion tracking. Therefore, they are highly suitable for indoor environments and improve the practical usability of IMU-based motion tracking in many clinical and biomedical applications.

To further advance the proposed methods, it should be evaluated if combining the rotation-based and the orientation-based constraint can increase the robustness and consistency of the joint axes estimates. Furthermore, introducing and evaluating metrics to quantify the estimation uncertainty and methods for automatic (re-)triggering of the calibration when suitable motions are detected are important next steps to increase the usability of the method.

## Figures and Tables

**Figure 1 sensors-22-09850-f001:**
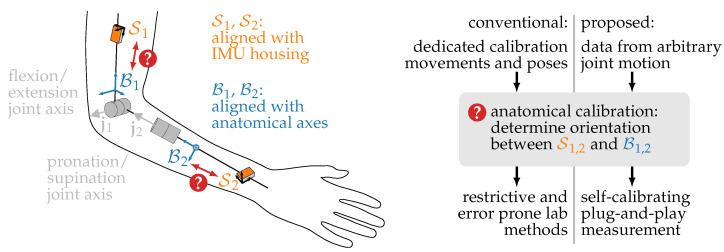
Anatomical calibration, also called sensor-to-segment calibration, is the task of determining how the IMUs are attached to the body segments. More precisely, the rotations between the IMU coordinate systems $S{}_{1,2}$, defined by the sensor housing, and the corresponding body segments $B{}_{1,2}$, determined by anatomical axes such as the joint axes $j_{1,2}$, have to be determined. Conventional methods rely on precisely defined calibration movements and poses, whereas the proposed methods use kinematic constraints to derive this information from arbitrary joint motion.

**Figure 2 sensors-22-09850-f002:**
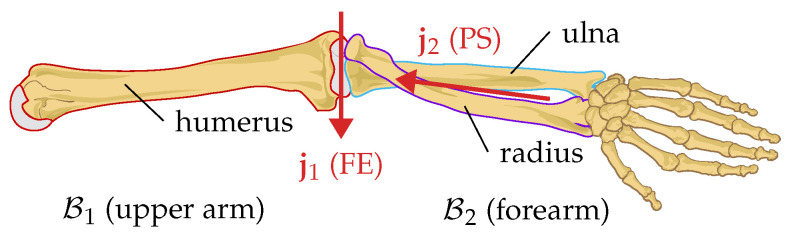
Anatomical model of the elbow joint. The humeroulnar joint is a hinge joint with the rotation axes $j_{1}$, allowing for flexion and extension (FE). The radioulnar joint also has one degree of freedom ($j_{2}$) and allows for pronation and supination (PS). In this contribution, we refer to the combined joint with two degrees of freedom as *elbow joint*.

**Figure 3 sensors-22-09850-f003:**
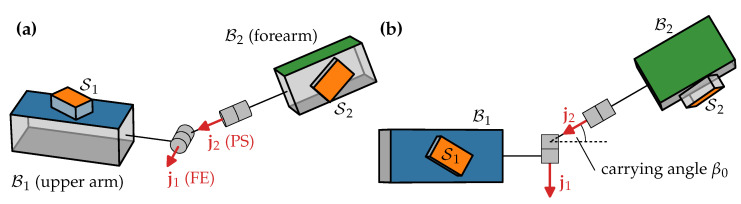
(**a**) Geometric kinematic model of the elbow joint. Inertial sensors $S{}_{1}$ and $S{}_{2}$ are placed in arbitrary orientation on the upper arm $B{}_{1}$ and forearm $B{}_{2}$. Upper arm and forearm are connected by two hinge joints that allow for FE ($j_{1}$) and PS ($j_{2}$). (**b**) View onto the $j_{1}$-$j_{2}$ plane. The fixed rotation between FE and PS is called *carrying angle*.

**Figure 4 sensors-22-09850-f004:**
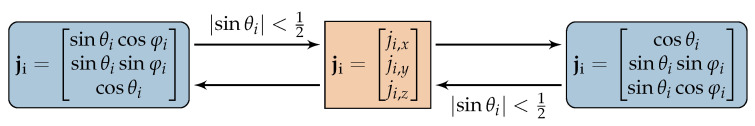
Two spherical parametrizations are used to represent the joint axes $j_{i}$, i=1,2, with two parameters each, $\theta_{i}$ and $\varphi_{i}$. To avoid the derivative becoming close to zero, we convert the respective axis to Cartesian coordinates and then to the other representation whenever $|\sin\theta_{i}|<0.5$.

**Figure 5 sensors-22-09850-f005:**
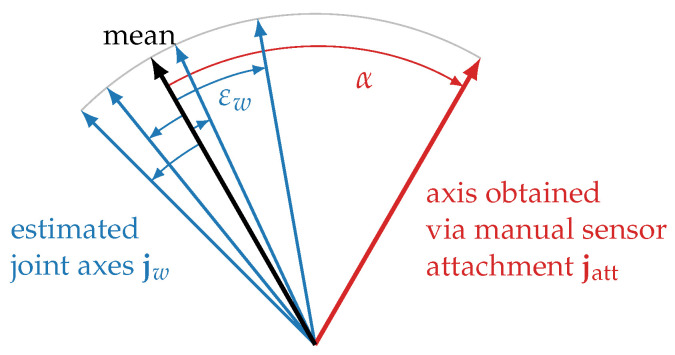
Variability angle $\varepsilon_{w}$ and misalignment angle α used to evaluate the axis estimation results. $\varepsilon_{i}$ is the angle between the estimated axis for a single window and the mean estimate. α is the angle between the mean estimate and the axis obtained by careful manual sensor attachment. For a good anatomical calibration method, $\varepsilon_{i}$ should be small, showing that the estimates are consistent, and α should be within 30°, showing that the estimates are plausible.

**Figure 6 sensors-22-09850-f006:**
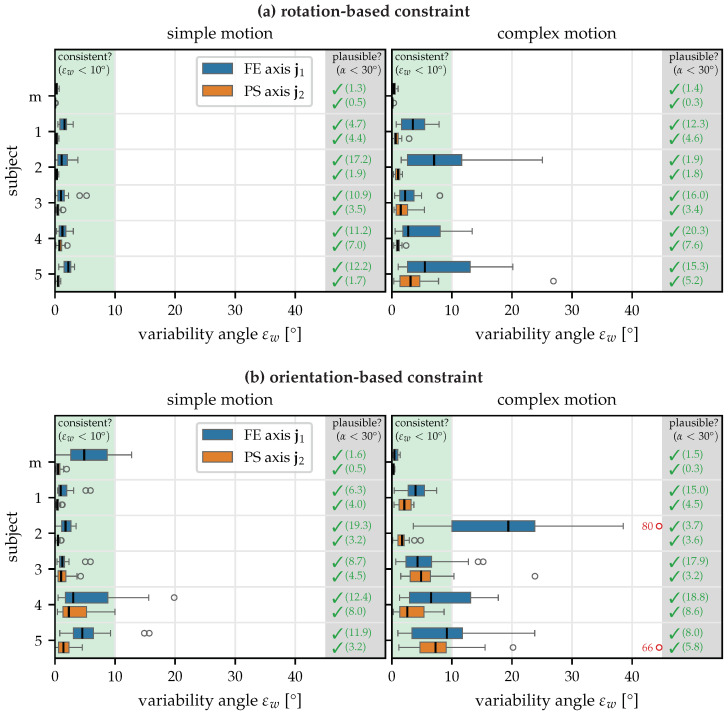
Consistency and plausibility results for the first experiment with the (**a**) rotation-based constraint and the (**b**) orientation-based constraint, for two motion types and for five human subjects and a mechanical joint (m). The proposed methods estimate plausible axes for all subjects and all motions. The rotation-based constraint yields more consistent estimates than the orientation-based constraint, and the simple motion leads to better results than the complex motion.

**Figure 7 sensors-22-09850-f007:**
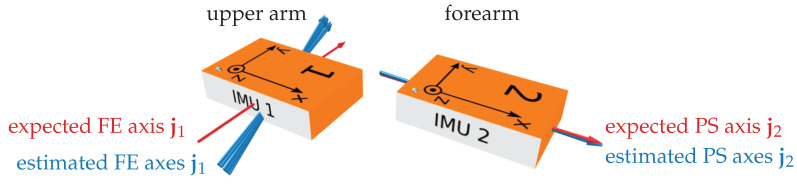
3D visualization of the estimation results for an exemplary trial (Subject 2, simple motion, rotation-based constraint). The joint axis estimates from all windows agree well (blue arrows). The PS axis agrees very well with the expected value (red arrow), while for the FE axis there is a misalignment of 17°, most likely due to imprecise manual sensor attachment.

**Figure 8 sensors-22-09850-f008:**
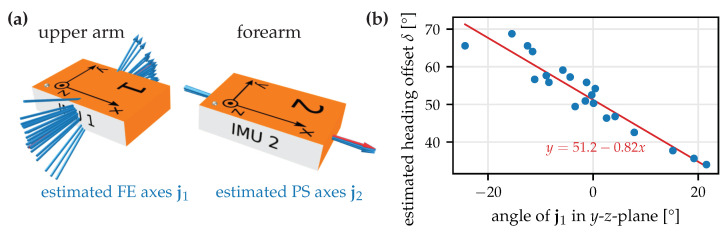
Investigation into the variability of the FE axis estimates (Subject 2, complex motion, rotation-based constraint). (**a**) 3D visualization of the axis estimates for all windows. (**b**) Plot of the estimated heading offset δ and the angle of the FE axis in the (approximately horizontal) *y*-*z*-plane of the upper arm IMU coordinate system. There is an obvious correlation, indicating that without sufficient upper arm movement, the kinematic constraint does not allow for distinguishing between a heading rotation of the joint axis and a heading offset between the sensor orientations.

**Figure 9 sensors-22-09850-f009:**
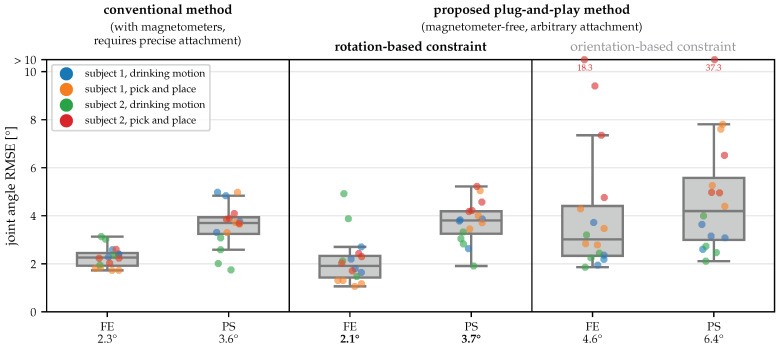
Joint angle estimation errors for all trials with a conventional 9D approach and with the proposed plug-and-play magnetometer-free methods, using OMC-based angles as ground truth. The numbers below the axis labels indicate the mean root mean square error (RMSE) for all 16 trials. The proposed method with the rotation-based constraint yields the same accuracy as the much more restrictive conventional 9D method.

**Figure 10 sensors-22-09850-f010:**
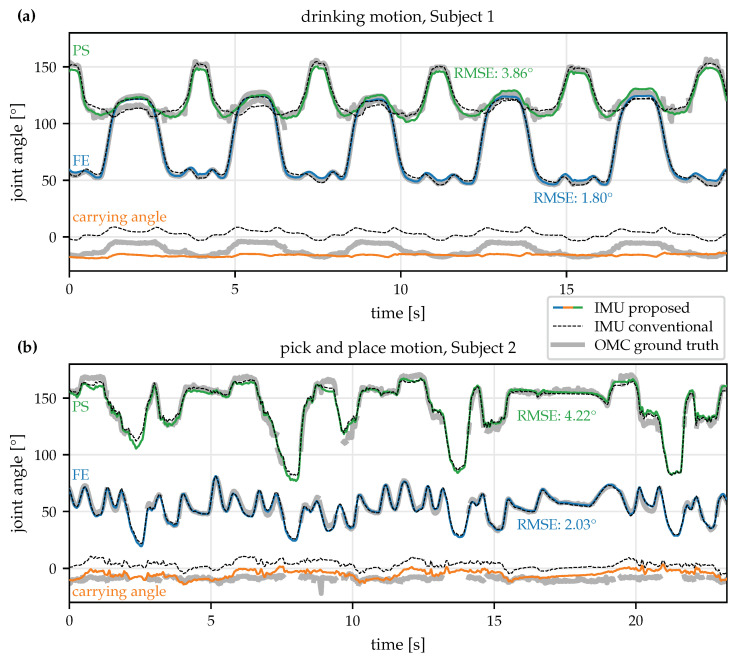
Joint angle trajectories for an exemplary (**a**) drinking and (**b**) pick-and-place trial obtained with the proposed IMU-based method (and the rotation-based constraint), the conventional 9D IMU-based approach, and the OMC ground truth. While being much less restrictive, the proposed method is able to obtain FE and PS joint angles that agree well with the angles obtained with the other two methods.

**Figure 11 sensors-22-09850-f011:**
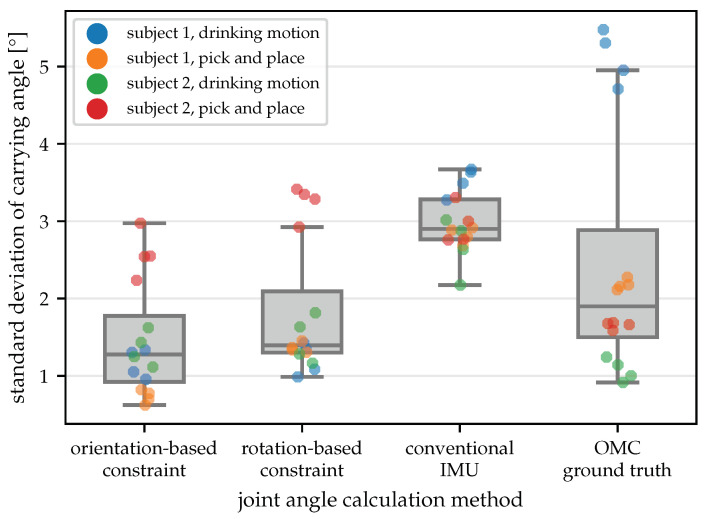
Standard deviation of the carrying angle for all trials with the different angle calculation methods. The proposed method induces the smallest variation in the assumed-to-be-constant carrying angle. This indicates that the estimated joint axes describe the functional motion axes better than the axes obtained via careful manual IMU placement (conventional IMU) and via placing markers on anatomical landmarks (OMC ground truth).

## Data Availability

The data presented in this study are available on request from the corresponding author.

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
