# Peer review of "Self-Calibrating Magnetometer-Free Inertial Motion Tracking of 2-DoF Joints"

_sensors, 2022, doi:10.3390/s22249850_

Round 1
Reviewer 1 Report
In this paper, the authors have proposed and demonstrated the efficacy of a new strategy for calibrating IMUs. The proposed method does not use the magnetometer signal, which can be unreliable, and instead exploits the fact that the carrying angle of the elbow joint is constant. In general, this is a high-quality paper with good motivation, sufficient technical detail, convincing results, and clear figures. The authors are particularly commended on having included results from a sensitivity analysis on the algorithm parameters, a topic that is very important in any computational study but is often ignored. The authors may wish to consider the following suggestions before proceeding with publication:
1. In several places, the authors refer to the "elbow joint" as a 2-DoF joint (e.g., lines 8, 45 and 129). Some explanation is provided around line 150, but even this explanation is somewhat confusing. For example, in the caption for Figure 2, the term "elbow joint" is used to mean two things: (a) the flexion/extension joint and (b) both flexion/extension and pronation/supination joints. Defining a term to mean two things may be confusing for readers. It might be clearer if, on line 45, the term "elbow joint" were enclosed in quotation marks followed by "(flexion/extension and pronation/supination)" so it is clear how the authors are using this term throughout their paper. In the caption for Figure 2, the term "elbow joint" could be used to refer to the 2-DoF joint but perhaps use the term "humeroulnar joint" to refer to the flexion/extension DoF on its own.
2. In Figure 2, "ulnar" should be "ulna".
3. The paragraph beginning on line 31 is only one sentence long and does not stand on its own since the word "this" in the phrase "this is often achieved" refers back to the previous paragraph. It would be clearer structurally if the sentence were moved to the end of the previous paragraph.
4. On line 43, the authors mention that "arbitrary joint motion" can be used for calibration. It might be appropriate to include here a mention of the very popular functional method for determining the hip joint center using marker-based motion capture data (e.g., see Piazza et al., "Accuracy of the functional method of hip joint center location: effects of limited motion and varied implementation", DOI 10.1016/S0021-9290(01)00052-5), to provide additional context.
5. The acronym "IOE" is defined on line 229 and only appears once more, on line 318. There does not appear to be any benefit to defining this acronym and the reader is unlikely to remember what it means. The list of abbreviations at the end of the paper is not particularly helpful. Perhaps simply write this phrase out again on line 318.
6. In Figure 4, would there not be a discontinuity in the angles and angular velocities when the switching occurs? Does this cause any numerical or other issues?
7. On line 305, the authors mention minimizing the sum of squares of the errors, which in Appendix B appears to minimize a multi-objective cost function consisting of the "rotation-based constraint" and "orientation-based constraint" (presumably, equations 17 and 18; referring to these equation numbers in the Appendix would be helpful). These errors have different units so it seems that they cannot actually be added. Also, presumably the answers would be different if the angles were measured in degrees vs. radians. Have the authors considered including weights, as is often done in multi-objective cost functions? Even if numerically equal to 1, weights can be helpful for canceling the different units.
8. On line 378, the authors should (must?) mention that ethics approval was obtained as is required when performing any human experiments (also on line 589). In addition to mentioning that the subjects were "healthy", presumably additional exclusion criteria were used and should be mentioned (the participants were presumably volunteers, not children, had no upper-limb injuries, etc.).
9. On line 384, the authors mention that the elbow was flexed "while keeping the arm in the frontal plane of the shoulder." It is quite uncomfortable to flex the elbow while keeping the arm in the frontal plane (i.e., with palms pointed laterally) so this is probably not what was intended. Perhaps a simple illustration or a clearer written description of the movement could be provided.
10. On line 403, perhaps define this angle as "the POSITIVE angle between two 3D vectors" or provide a mathematical definition that includes absolute value bars. Presumably this is what was intended, not a signed angle.
11. In Figure 10(b), the "ground truth" data are missing in several places. Please mention why these data are missing, particularly given that the data are missing at points during the motion where comparison to ground truth would be helpful.
12. On line 529, which subject were the "exemplary trials" obtained from and what were the errors corresponding to these trials? Providing this information would help the reader interpret these selected trials in the context of the population-level data presented in Figure 9.
Reviewer 2 Report
This study proposes an I2S calibration method for 2-DoF hinge joints without using magnetometers or sensor orientation information calculated based on the magnetometers. Section 1 presents the background and purpose of this study, touching on the reasons why magnetometers should not be used; Section 2 introduces existing calibration methods and explains the significance of I2S calibration, even if only for hinge joints; Section 3 describes the skeletal model used; and Section 4 describes the proposed method with detailed equations. In Section 5, the validity of the proposed method is confirmed by verifying the proposed method using two different methods. Although there are still some issues as indicated in Chapter 6, the research and the quality of the paper are both sufficient, and we recommend that the paper be accepted by Sensors. The following are minor comments.- Please clarify the two types of constraints in the abstract and in Section 1.
- Section 4 describes the computational procedure in detail, but I would like to see a rough description of the method at the beginning of the section to improve readability.
- Section 2 is a good summary of general I2S calibration methods. On the other hand, there is a lack of reference to non-magnetic methods, including methods using machine learning, which is a recent trend. There are also few references to calibration methods limited to hinge joints, as in this study. I would appreciate a description of existing research that sets up a problem similar to this study.
